# Landslide Detection Based on ResU-Net with Transformer and CBAM Embedded: Two Examples with Geologically Different Environments

**Zhiqiang Yang** [1,2] , **Chong Xu** [1,2,*] **and Lei Li** [1,2,3]

1   National Institute of Natural Hazards, Ministry of Emergency Management, Beijing 100085, China; yangzhiqiang20@mails.ucas.ac.cn (Z.Y.); leil_cugb@126.com (L.L.)
2   Key Laboratory of Compound and Chained Natural Hazards Dynamics, Ministry of Emergency Management of China, Beijing 100085, China
3   School of Engineering and Technology, China University of Geosciences (Beijing), Beijing 100083, China
*   Correspondence: chongxu@ninhm.ac.cn

**Abstract:** An efficient method of landslide detection can provide basic scientific data for emergency command and landslide susceptibility mapping. Compared to a traditional landslide detection approach, convolutional neural networks (CNN) have been proven to have powerful capabilities in reducing the time consumed for selecting the appropriate features for landslides. Currently, the success of transformers in natural language processing (NLP) demonstrates the strength of self-attention in global semantic information acquisition. How to effectively integrate transformers into CNN, alleviate the limitation of the receptive field, and improve the model generation are hot topics in remote sensing image processing based on deep learning (DL). Inspired by the vision transformer (ViT), this paper first attempts to integrate a transformer into ResU-Net for landslide detection tasks with small datasets, aiming to enhance the network ability in modelling the global context of feature maps and drive the model to recognize landslides with a small dataset. Besides, a spatial and channel attention module was introduced into the decoder to effectually suppress the noise in the feature maps from the convolution and transformer. By selecting two landslide datasets with different geological characteristics, the feasibility of the proposed model was validated. Finally, the standard ResU-Net was chosen as the benchmark to evaluate the proposed model rationality. The results indicated that the proposed model obtained the highest mIoU and F1-score in both datasets, demonstrating that the ResU-Net with a transformer embedded can be used as a robust landslide detection method and thus realize the generation of accurate regional landslide inventory and emergency rescue.

**Keywords:** landslide detection; deep learning; transformer; ResU-Net; CBAM; self-attention

## 1. Introduction

As one of the most critical types of natural hazards, landslides are triggered by various external factors in most cases, including earthquakes, rainfall, variation of water level storms, and river erosion [1]. Once landslides develop into geological hazards, there is a great potential to cause devastating damage to natural structures and infrastructure, leading to human casualties and property damage [2]. A detailed landslide inventory records the location, date, and type of landslide in a given area, which can serve as a basis for further investigation and provide geologists with scientific data used to assess landslide risk [3,4]. Additionally, studies have shown that the legacy effect may lead to a higher chance of landslides occurring on previous landslide paths in the next decade. Therefore, generating landslide inventory is the first step in hazards management and susceptibility evaluation [5,6]. The development of advanced earth observation techniques provides unique opportunities for the comprehensive assessment of disaster losses [7], thus massive amounts of satellite images are gradually replacing field surveys as a low-cost resource

for building a landslide database. As a consequence, how to detect landslides rapidly in large-scale remote sensing datasets is of great significance for disaster mitigation [8–10].

The main landslide detection methods based on remote sensing include (1) Pixel-based, (2) Object-oriented, (3) Machine Learning, and (4) Deep Learning. Among them, the adjacent pixels are not taken into account in the pixel-based method which mainly extracts landslides by comparing the images' intensity or the band difference between two-phase images. For example, Nichol and Wong [11] attempted to extract landslide data using single-band difference, band ratio, and post-classification comparison. Zhao et al. [12] selected the normalized difference vegetation index (NDVI), enhanced vegetation index, and soil-adjusted vegetation index as features to classify the landslides. Mondini et al. [13] detected the rainfall-induced landslides by adopting four change detection. However, the pixel-based method has rigorous requirements for image acquisition and processing. Besides, the improved resolution and differences in the distribution of landslides may introduce pepper noise [3]. The object-oriented approach is a knowledge-driven approach (OOA), in compliance with the assumption that pixels with neighbors belong to the same class and treat landslides as collections of pixels [14]. For example, Lu et al. [15] used temporal principal component analysis, spectral angle mapper, Reed-Xiaoli detection, and textural analysis to derive object features as segmentation metrics and performed multi-level segmentation through a multiscale optimization approach. Rau et al. [16] adopted a multilevel segmentation and hierarchical semantic network for landslide detection. Eeck-haut et al. [17] only used derivatives of LiDAR for landslide detection in high vegetation cover areas. Although OOA takes the shape and texture properties of landslides into consideration to refine the result, the parameters, such as segmentation scale and features, are subjectively optimized by the user, resulting in a time-consuming process and low automation [18,19].

Machine learning [ML] has excellent potential in classification and capacity to handle data of high dimensionality and to map classes with very complex characteristics. Commonly used methods include support vector machines (SVM), decision trees (DTs), random forest (RF), and artificial neural networks (ANN) [20]. Among them, [21,22] have demonstrated that RF exhibited stable portability on new data compared to SVM and ANN. Piralilou et al. [23] proposed a landslide detection method connecting the result of OOA with logistic regression, multilayer perceptron, and RF by the Dempster-Shafer theory. Although machine learning is widely applied in the field of remote sensing, high-dimension data may overfit models or introduce noise, causing performance deterioration. Hence, uncertainties analysis in ML-based models is an indispensable part to select important variables, evaluate the sensibility of the input data and the generalization of the new samples [24,25]. For example, Adnan et al. [26] and Rossi et al. [27] combined the landslide susceptibility maps generated by models to address the uncertainty from the single model.

As a hot field in computer vision, deep learning (DL) has shown outstanding performance in image classification and segmentation [28–30]. In terms of landslide susceptibility mapping (LSM), Dao et al. [31] probed into the performance of the DL model which significantly improved the ability to predict future landslides compared to three ML models. Similarly, by introducing nine controlling factors related to landslides, Bui et al. [32] investigated the capability of the DL model in LSM compared to SVM, C4.5 and RF, and the results indicated that the DL model had a more stable classification performance. On the other hand, CNN equipped with VGGNet, ResNet, and DenseNet has been widely applied in landslide detection. For example, Cai et al. [33] and Liu et al. [34] incorporated controlling factors into samples to test the feasibility of Dense-Net in landslide extraction, while Nava et al. [35] explored the performance of the network when topographic factors were fused into SAR images. Furthermore, Ghorbanzadeh et al. [36] attempted to fuse landslide detection results generated from CNN trained with different datasets through the Dempster-Shafer model. Regarding ResU-Net, Prakash et al. [37] first explored the potential of U-Net with ResNet34 in landslide mapping, demonstrating the utility of deep learning based on EO data for regional landslide mapping. Furthermore, Qi et al. [38]

proved that the U-Net with ResNet50 can improve the performance of the model on rainfall-induced landslide detection. Ghorbanzadeh et al. [39] firstly used the free Sentinel-2 data in landslide identification by evaluating the performance of U-Net and ResU-Net in three different landslide areas, the result indicated that ResU-Net obtained the highest F1-score. To alleviate the lack of generalization of the model caused by various landslide morphologies, Yi et al. [18] introduced an attention mechanism to assign weights for important feature maps in ResU-Net, the F1-score on the proposed model was improved by 7% compared to ResU-Net. Recently, Ghorbanzadeh et al. [40] designed a novel approach for landslide extraction. By integrating an OBIA-based model and ResU-Net, the proposed model addressed the fuzzy landslide boundaries problem by enhancing and refining the results generated by ResU-Net and possessed higher precision, recall, and F1-score. Besides, Ghorbanzadeh et al. [41] created a public landslide dataset for the landslide detection community, denoted as Landslide4Sense, which contains 3799 images fusing optical bands from Sentinel-2 with DEM and slope. Overall, CNN possesses robustness and scalability, and features are automatically extracted through hierarchical structure and convolution, avoiding the involvement of excessive domain knowledge [42]. However, studies have shown that the actual receptive field in CNN is much smaller than the theoretical size, which is not conducive to making full use of global information [43].

The proposal of a transformer provides an efficient parallel network to natural language processing (NLP) [44,45], and its architecture dominated by multi-head self-attention (MSA) abandons recursion and convolution. Different from CNN that expands the receptive field by stacking convolutions, MSA captures global contextual information at the beginning of feature extraction and establishes long-range dependencies on the target [46,47]. It has been confirmed that the inherent inductive biases, such as transition equivalence and locality, are excluded in the transformer. Among them, the former well respects the nature of the imaging process while the latter controls the model complexity by sharing parameters across space [48]. Nevertheless, the transformer still outperforms CNN when pretraining to a transformer with a large dataset [49,50].

The outstanding performance of the vision transformer (ViT) on classification confirmed that a transformer can be directly used in computer vision. For example, Bazi et al. [51] verified that ViT achieved high classification accuracy on four datasets. Reedha et al. [52] introduced ViT into vegetation classification with a small dataset, achieving higher F1-score compared to EfficienNet and ResNet. Recently, research by Park and Kim [53] showed that MSA will flatten the loss value, guiding the model to converge along a smooth trajectory. In addition, MSA is equivalent to a low-pass filter for aggregating feature maps, while convolution is similar to a high-pass filter, making feature maps more diverse, such phenomenon demonstrated that MSA and convolution complement each other when capturing features. As a consequence, numerous studies aiming to combine the CNN and transformer have appeared, for instance, Deng et al. [54] proposed CTNet, which introduced the ViT to explore global semantic features and employed CNN to compensate for the loss of local structural information. Zheng et al. [45] designed a segmentation network called SETR to model the global context of images based on the pure transformer. The proposed model could obtain excellent results only by combining with a simple decoder. Horry et al. [55] introduced a highly scalable and accurate ViT model into a low-parameter CNN for land use classification, aiming to cope with the growing amount of satellite data. Similarly, Xu et al. [56] proposed an end-to-end network, called ET-GSNet, to improve the generalization of the different tasks by knowledge distillation. Besides, transformer and convolution-based SAR image despeckling networks were proposed by Perera et al. [57] that also possessed superior performance compared to the pure CNN.

Given the strength of transformer in global context modeling, and that, as far as we know, there has been no research evaluating its potential in landslide detection. In this study, a segmentation network incorporating a transformer in ResU-Net proposed by Chen et al. [58] was selected to validate its effectiveness in landslide detection with small datasets, and the pre-trained weight on Imagenet21K was introduced to accelerate model

convergence. To better fuse the feature maps from transformer and CNN, a spatial and channel attention module was embedded into the decoder. Finally, two landslide datasets with different characteristics were chosen to confirm the potential of the proposed model thoroughly.

## 2. Study Area

### 2.1. Iburi Dataset

The Iburi dataset is located in Iburi, Hokkaido, Japan, with a total area of 733.06 km². The landforms in this area are mainly hills with moderate slopes and an altitude range from 0–624 m, as shown in Figure 1a. According to JMA: Hokkaido's annual average temperature is 6–10 °C, and the annual precipitation is 800–1200 mm. At 3:08 local time on 6 September 2018, an Mw 6.6 earthquake struck this region, inducing a large number of coseismic landslides, most of which were shallow soil landslides, as shown in Figure 1b [59]. The Iburi dataset was interpreted by our team based on the pre-and post-earthquake Planet 3 m resolution images. To ensure the correct interpretation, we combined the high-resolution drone images provided by the Geospatial Information Authority of Japan, and finally obtained 9293 landslides.

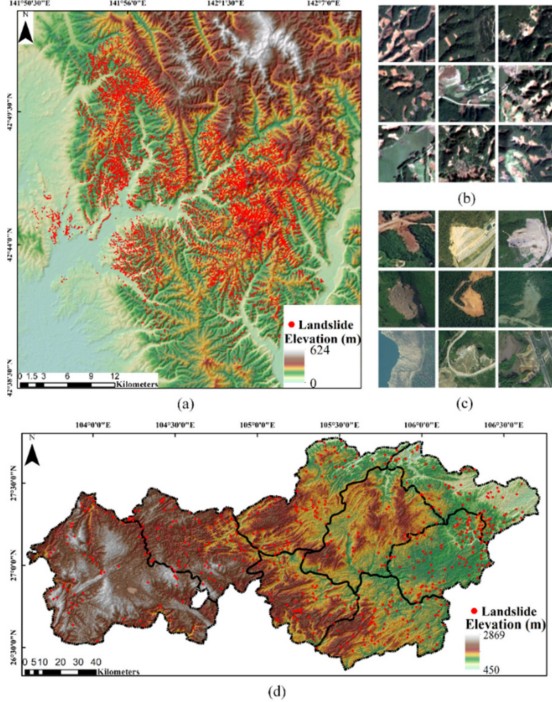

**Figure 1.** The geographical distribution of landslide datasets. (**a**) is the landslide distribution and topography in the Iburi district. (**b**) is the detailed landslide shapes in the Iburi dataset, the majority of landslides are continuous small- and medium-sized shallow detrital slides. (**c**) is the detailed landslide shapes in the Bijie dataset. An image contains a landslide. The size of both datasets is 224 × 224 pixels, and the resolutions are 3 m and 0.8 m respectively. (**d**) is the landslide distribution and topography in Bijie city.

### 2.2. Bijie Dataset

Considering the inadequacy of landslide-triggered landslide samples, a public landslide dataset, denoted as the Bijie dataset, was selected to evaluate the generalization of the proposed model. The Bijie dataset mainly consists of 770 landslide images which are all located in Bijie City, Guizhou Province, China. All landslides were extracted from TripleSat images with 0.8 m resolution based on visual interpretation and the majority of landslides were induced by rainfalls, earthquakes, and human activities [60,61], as shown in Figure 1c.

Bijie City is situated in the transitional slope zone from the Tibet Plateau to the eastern hills with an altitude of 450–2869 m. Fragile geological environment, unstable geology, undulating terrain and abundant rainfall (annual average rainfall is 849~1399 mm) make it a high landslide-prone area [62], as shown in Figure 1d. In addition, frequent extreme weather has exacerbated the suddenness of geological disasters in this region. Therefore, there is a strong demand for automated and rapid landslide detection approaches for emergency rescue and risk evaluation in this region.

## 3. Methodology

### 3.1. Self-Attention and Multi-Head Attention

In the transformer, assuming that there is a processed sequence different from the CNN that generates the attention map through convolution and pooling [63], the goal of self-attention is to capture the interaction amongst all n entities by encoding each entity in terms of the global contextual information [47,52]. The implementation of the self-attention can be briefly described as a process of mapping queries ($Q$), keys ($K$), and values ($V$) to outputs [46], where queries, keys, and values are generated by projecting the input sequence $X = (x_1, x_1, \cdots x_n) \in \Re^{n \times d}$ to three learnable weight matrices ($W^q \in \Re^{n \times d_q}$, $W^k \in \Re^{n \times d_k}$, $W^v \in \Re^{n \times d_v}$) as shown in Equation (1). The specific attention score is computed by scaled dot-product-attention [46]. As shown in Figure 2a, the attention score is acquired through the following steps: (1) $Q$ multiplies the transposed $K$ (2) divided by the $\sqrt{d_k}$ to reduce the weighting range (3) passes through the softmax function to obtain the final score. The detailed calculation equation of the attention score can be expressed as follows:

$$Q = X \times W^q, K = X \times W^k, V = X \times W^v \tag{1}$$

$$\text{Attention}(Q, K, V) = \text{softmax}(\frac{Q \cdot K^T}{\sqrt{d_k}}) \cdot V \tag{2}$$

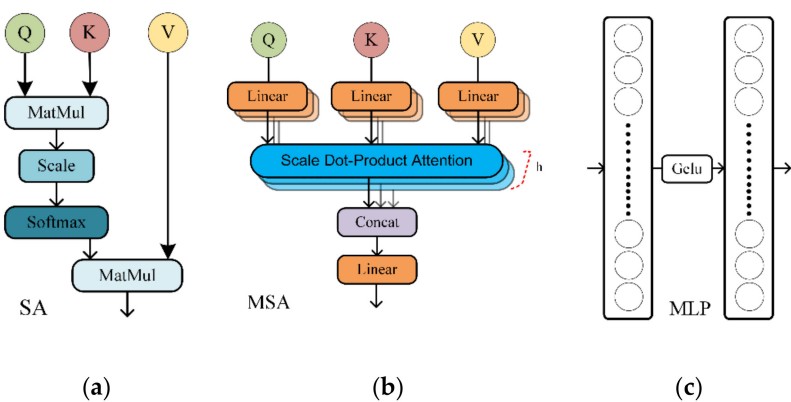

**Figure 2.** The structure of Self Attention (**a**), Multi-Head Self Attention (**b**), Multi-Layer Perceptron (**c**). (**a**) represents the standard calculation of the self-attention score, (**b**) is the calculation of self-attention score in transformer. (**c**) is a part of transformer.

Compared to CNN, self-attention increases the receptive field without increasing the computational cost associated with kernel sizes. In addition, self-attention is invariant to permutations and changes in the number of input points, hence it can easily operate on irregular inputs instead of the standard convolution that requires a grid structure [52,64]. In transformers, self-attention is extended to multi-head self-attention by computing h self-attention operations in parallel, in which h is called the head. Then, they are concatenated and projected to generate the final attention weights, as shown in Figure 2b. The purpose of using multi-head attention is to obtain semantic information in different subspaces so that

the network can focus on multiple types of features, which is similar to the multi-channel in CNN. The detailed equation of MSA can be expressed as follows:

$$\text{MSA}(Q, K, V) = \text{Concat}(\text{head}_1, \cdots \text{head}_h) \cdot W^v \tag{3}$$

$$\text{head}_i = \text{Attention}(QW_i^q, KW_i^k, VW_i^v) \tag{4}$$

### 3.2. Vision Transformer

The vision transformer aims to introduce 3D pictures into the transformer structure [49]. It avoids the limitations in CNN that global information is obtained by expanding the receptive field layer-by-layer and helps the network model the global context. The complete ViT structure consists of an embedding layer, an encoder, and a head [51], as shown in Figure 3a.

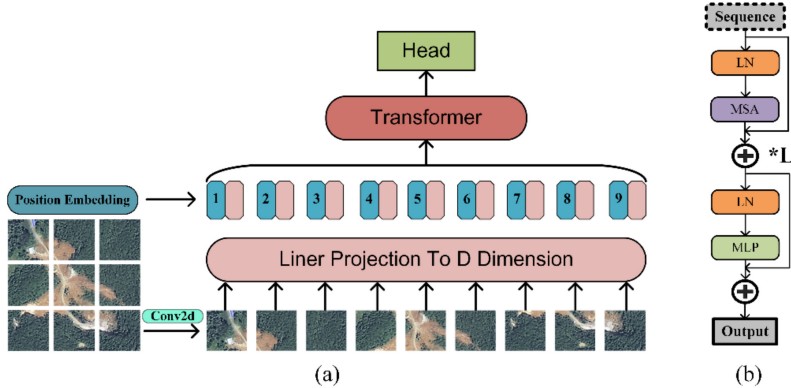

(a)                                                                    (b)

**Figure 3.** The structure of vision transformer (**a**), transformer (**b**). In (**a**), conv2d aims to reshape the images into nine parts. Position Embedding gives patch the location information. In (**b**), *L represents the layer number, default is 12.

The role of the embedding layer is to complete the conversion of 3D images to sequences to fit the input structure of the transformer. The detailed steps are listed as follows: Given a 3D image, it is first reshaped into a set of non-overlapping and flat 2D patches $x_n \in \Re^{n \times (P^2 \cdot C)}$, where $H$, $W$, $C$ represent the height, width, and channel of the images, $P$ represents the resolution of the patches, and n is the number of embedded patches. Subsequently, the patches are projected to the D dimension by a learnable embedding matrix $E \in \Re^{(P^2 \cdot C) \times d}$ [56]. Up to this point, an image is transformed into a sequence consisting of n patches which can also be called tokens. In the classification task (original article), an equally shaped and learnable patch is embedded into the sequence as the final classification representative of the model [54], while in the semantic segmentation task, this is not needed. Although global modeling facilitates the extraction of long-range semantic information, the lack of convolution makes it difficult for the model to capture positional features. Thus, features capable of representing relative or absolute positional information need to be embedded in the sequence [46,49], in ViT, a learnable 1D parameter with the same size as the sequence is summed to form the positional information. The above steps can be summarized as the following equation:

$$X = [x_1 E; x_2 E; \cdots; x_n E] + E_{pos} \tag{5}$$

The transformer is the encoder of ViT, it consists of L layers with the same structure. In each layer, MSA and MLP (multilayer perceptron) are the main components aiming to aggregate important tokens in the sequence [53], as shown in Figure 2b,c. The specific composition is shown in Figure 3b. First, the sequence normalized through the layer normalization is inputted into the MSA to calculate the importance weights among all

tokens, and the original information is preserved by using the residual structure. Among them, MSA extracts feature information from different subspaces and then concatenates the result from h self-attention head, which enables the model to selectively assign more focus to certain positions of the image, and also helps the model to learn features faster and reduce the training cost [49,52]. Then, the obtained results are sequentially passed through an LN and an MLP, and the same residual structure is introduced to generate the final semantic information extracted by this layer.

### 3.3. Convolution Block Attention Module

Since the combination of ResNet-50 and a transformer was selected as the backbone and the pre-trained weights need to be loaded to accelerate the convergence, changing the encoder structure may be inappropriate and inconsistent with the aim of this work. In [65], the attention module Squeeze-and-Excitation (SE) was embedded into the segmentation head to selectively enhance the weight of important features and reduce the weight of less concerning features in channels. Inspired by this, we introduced an attention mechanism in skip connection for the feature maps generated by CNN and the transformer, driving the decoder to focus on more important information in both the global and local feature maps and suppress the unnecessary features. The Convolutional Block Attention Module (CBAM) is a simple and effective attention module that can be seamlessly integrated into any CNN with a negligible number of parameters [66]. As shown in Figure 4, the module derives attention weights along the space and channel, respectively, and then multiplies the weights with the feature maps for adaptive refinement [53].

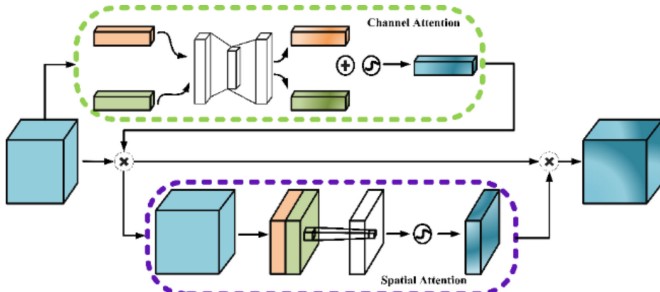

**Figure 4.** The structure of the Convolutional Block Attention Module. The upper represents the channel attention and the lower represents the spatial attention, the feature maps first pass through the upper and then the lower.

The implementation of CBAM is depicted as follows: Given a feature map $F \in \Re^{H \times W \times C}$ from the encoder, the CBAM derives a one-dimensional channel weight map $C \in \Re^{1 \times 1 \times C}$ and a two-dimensional spatial attention map $S \in \Re^{H \times W \times 1}$ by convolution and pooling. In the channel attention module, the channel weights are described as Equation (6). Specifically, the spatial information is compressed and aggregated into two $C \times 1 \times 1$ feature maps by global average pooling and global max pooling, then they are forwarded to a multilayer perceptron (MLP) and summed. Finally, the sigmoid function is applied to generate the final channel attention map.

$$C = \text{Sigmoid}(\text{MLP}(\text{Maxpool}(F)) + \text{MLP}(\text{Avgpool}(F))) \tag{6}$$

$$S = \text{Sigmoid}(\text{Conv}(\text{Concat}(\text{Maxpool}(F), \text{Avgpool}(F)))) \tag{7}$$

In the spatial attention module, the weight values are calculated by Equation (7). In detail, $F \in \Re^{H \times W \times C}$ are firstly fed into global max pooling and global average pooling along the channel axis to generate two $1 \times H \times W$ feature maps, and then are concatenated to generate an $2 \times H \times W$ feature map, in which a convolution operation is applied to

reduce the dimensions to $1 \times H \times W$. Finally, the spatial attention map is generated by a sigmoid function. The overall procedure of CBAM is written as:

$$FP_{\text{refine}} = S(C(FP)) \tag{8}$$

### 3.4. Attention TransU-Net

ResU-Net [67,68], as a classical network in semantic segmentation, has achieved great success in landslide detection tasks [34,38,39]. It has been demonstrated that the skip connection in U-Net effectively alleviates the detail loss problem due to convolution, and the residual structure prevents the vanishing gradient when mining deeper information [38]. To reduce the limitations of convolution in long-range information extraction, Chen et al. [58] designed a medical image segmentation network called TransUnet, introducing a transformer to model the global context of feature maps generated from ResU-Net, and the proposed model achieved higher accuracy compared to ResU-Net.

In this work, we built a modified TransUnet referring to [58], called CTransUnet (C represents CBAM), that suited the purpose of landslide detection and embedded the attention module proposed in Section 3.2. The proposed model consists of an encoder and decoder, the detailed structure is shown in Figure 5b. In the encoder, ResNet-50 is used to extract feature maps with the different receptive field, and then the introduced transformer aims to model the global context of the feature maps. Unlike the standard ResNet network, the Conv is converted to StdConv, same as in [49], and the batch normalization is replaced with group normalization which is more suitable for the transformer. Besides, the third and fourth stages in ResNet-50 are merged as shown in Figure 5a. The input image is finally converted to three feature maps in ResNet, in which all feature maps provide high-resolution information for the decoder, and the deepest feature map is selected as input for the transformer. Finally, the feature maps generated by transformer $X \in \Re^{n \times d}$ are reshaped into $\frac{H}{P} \times \frac{W}{P} \times d$. In the decoder, the feature maps are resampled to the full resolution by a cascaded upsampler which consists of multiple upsamling blocks. Each block is composed of a $2\times$ upsamling operator, a $3 \times 3$ convolution operation, and a ReLu activation function successively [58]. In addition, after concatenating the feature maps at each layer, the CBAM module is embedded to further calculate weights for all features.

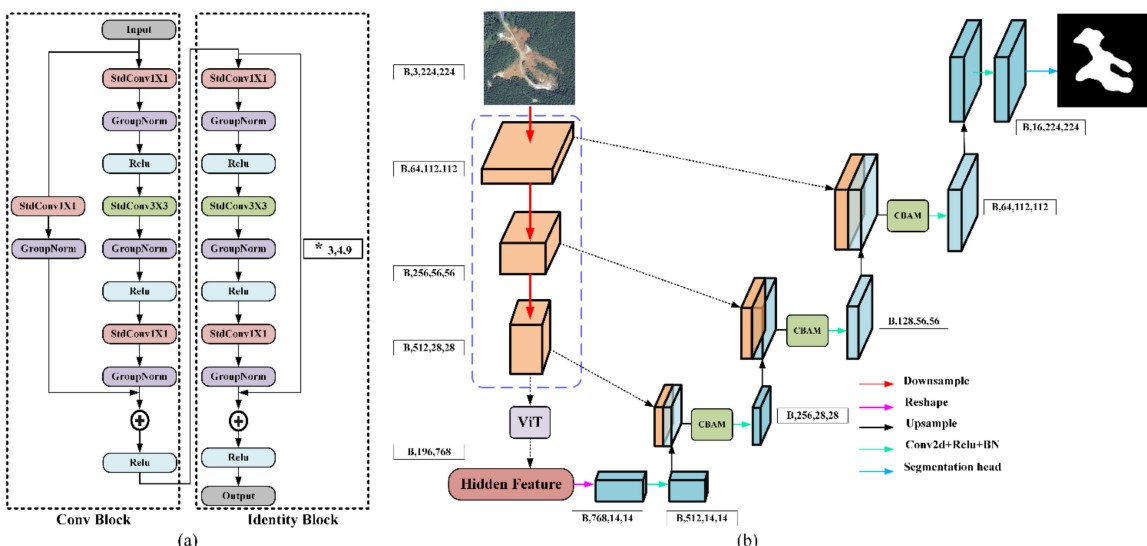

**Figure 5.** The structure of modified ResNet50 (**a**), CTansUnet (**b**). *3,4,9 represents the cycles of blocks in each stage of ResNet-50.

### 3.5. Evaluation

To evaluate the performance of the proposed model, the standard ResU-Net was selected as the benchmark in the experiments. In addition, we adopted five commonly used metrics to evaluate quantitatively, which include precision, recall, F1-score, IoU, and mIoU. Among them, precision and recall are defined as the following equations:

$$\text{Precision} = \frac{TP}{TP + FP} \tag{9}$$

$$\text{Recall} = \frac{TP}{TP + FN} \tag{10}$$

where *TP*, *FP*, and *FN* represent the true positive, false positive, and false negative respectively, as shown in Table 1:

**Table 1.** Confusion matrix for binary classification, 1 represents landslide pixel, 0 represents background pixel.

|  | **Prediction False** | **Prediction Truth** |
|---|---|---|
| **Ground False** | *TN* | *FP* |
| **Ground Truth** | *FN* | *TP* |

F1-score is the harmonic mean of precision and recall [34] as written in Equation (11), it combines the capabilities of the two metrics and can better describe the quality of classification. IoU (Intersection over Union) and mIoU (mean IoU) are often used to evaluate the ability of the model in semantic segmentation [21]. The former describes the ratio of the intersection and union set of the positive results and the real results, and is computed by Equation (12), the latter represents the averaged value of summed IoU of each class.

$$\text{F1} - \text{score} = \frac{2 * \text{precision} * \text{recall}}{\text{precision} + \text{recall}} = \frac{2 * TP}{2 * TP + FP + FN} \tag{11}$$

$$\text{IoU} = \frac{\text{Predict} \cap \text{Target}}{\text{Predict} \cup \text{Target}} = \frac{TP}{TP + FP + FN} \tag{12}$$

### 3.6. Dataset Preparation

In both datasets, a portion of the samples which were not involved in the training and validation process were selected as new data to assess the predictive power of both models [24,25]. For the 770 landslide images in the Bijie dataset, 70% were randomly selected for training with 20% for validation and 10% for testing. Considering the data volume and the distribution characteristics of the Iburi earthquake-induced landslides, a total of 296 images were cropped as the samples, of which 80% were randomly selected as training data and 20% as validation. Besides, 62 images were cropped for testing. The detailed parameters are shown in Table 2. To more perfectly match the pre-training weights, all samples were resized to 224 × 224 with RGB channels.

**Table 2.** Information of the datasets.

| Dataset | Size | Band | Train/Val/Test | Resolution |
|---|---|---|---|---|
| Bijie | 224 × 224 | RGB | 525/150/75 | 0.8 m |
| Iburi |  |  | 237/59/62 | 3 m |

Finally, seven types of data augmentation strategies, such as salty noise, rotation, and flip, were processed for all samples to prevent overfitting and increase generalization referring to the previous studies [50,69], as shown in Figure 6.

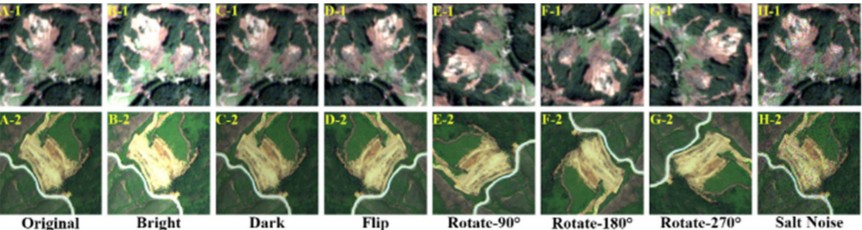

**Figure 6.** Data augmentation. *-1 represents the Iburi dataset, *-2 represents the Bijie dataset. All converts are implemented through OpenCV. * represents A–H.

### 3.7. Experimental Settings

For both encoders, the pre-trained weights on the ImageNet dataset [49] were used to accelerate downstream model convergence. We conducted three experiments in total. In the first set of experiments, the benchmark and the proposed model were trained separately using two datasets and then the performance was evaluated on the test datasets. Considering the parameter volume in ViT, we used base-level transformer parameters (i.e., patch size was set to 16, sequence length was 196, dimension of embedding patches was set to 768, dimension of the fully connected layer was set to 3072, head and layer were set to 12). In the second set of experiments, we evaluated the impact of data augmentation on both models. Finally, the influence of CBAM was evaluated to validate its potential in optimizing the decoder to process features maps.

In the training process, the trial-and-error method was adopted to select appropriate parameters. In detail, the SGD was selected as the optimizer with the momentum set to 0.9 and the weight decay set to 0.0001, the batch size was 48, and the drop-out in the transformer was set to 0.5 and 0.2 for the Iburi and Bijie datasets, respectively, the initial learning rate was set to 0.1, followed by a reduction proportion of 3/10 for every 20 epochs. Total epoch was set to 60. Considering the proportion of positive and negative pixels, the cross-entropy loss and dice loss were integrated to update the weight of the network, the former was less computational and helped the network to fit stably, while the latter mitigated the sample imbalance [70]. The loss function is expressed by the following equations:

$$L_{\text{total}} = 1 * L_{\text{ce}} + 0.5 * L_{\text{dice}} \tag{13}$$

$$L_{\text{ce}} = -(\hat{y} \cdot \log(y) + (1 - \hat{y}) \cdot \log(1 - y)) \tag{14}$$

$$L_{\text{dice}} = 1 - \frac{2 \cdot |\hat{y} \cap y| + \varsigma}{|\hat{y}| + |y| + \varsigma} \tag{15}$$

In Equation (14), $\hat{y}$ represents the label (1 is landslide, 0 is no-landslide), $y$ is the probability predicted by the model, while in Equation (15), $y$ denotes the predicted class. $\varsigma$ represents the smooth coefficient, aiming to avoid that divisor is 0.

The codes were implemented on Pytorch 1.10.1 and all experiments were conducted on a Dell graphics workstation with the following specification: Central processing Unit: Intel Xeon Gold 6238@2.10 GHz, a RAM of 512 GB, and two NVIDIA Quadro RTX 6000 (24 GB).

## 4. Results

### 4.1. Result Comparison

The proposed model and ResU-Net were trained with the same dataset for comparison. Besides, both models were initialized with pre-trained weights, and the results in test samples were evaluated based on ground truth. Tables 3 and 4 clearly illustrated that the proposed model had the best performance in two datasets compared to the ResU-Net in terms of the Recall, F1-score, IoU, and mIoU, demonstrating the segmentation potential of the proposed model on both datasets. Specifically, for the Bijie dataset, all metrics improved in the proposed model and the F1-score improved by 1.2%, reaching 87.73%, and mIoU improved by 1.04%, reaching 87.91%. For the Iburi dataset, the F1-score of the proposed

model improved by 1.84% and mIoU improved by 0.99%. Regarding precision, we found that the proposed model only increased by 0.39% in the Bijie dataset while it decreased by 1.15% in the Iburi dataset. It is worthwhile to mention that recall and IoU in both datasets improved considerably compared to precision and mIoU, the former indicated that the proposed model was more effective in suppressing false negatives in semantic segmentation of landslides, while the latter proved that the proposed model improved apparently the recognition rate of positive samples. Overall, the proposed model had superior performance compared to ResU-Net.

**Table 3.** Performance of models on the Bijie dataset. The bold means the variation between the proposed model and ResU-Net.

| Model | Precision | Recall | F1-Score | IoU | mIoU |
|---|---|---|---|---|---|
| Proposed | 87.24 (↑**0.39**) | 88.23 (↑**2.01**) | 87.73 (↑**1.2**) | 78.15 (↑**1.88**) | 87.91 (↑**1.04**) |
| ResU-Net | 86.85 | 86.22 | 86.53 | 76.27 | 86.87 |

**Table 4.** Performance of models on the Iburi dataset. The bold means the variation between the proposed model and ResU-Net.

| Model | Precision | Recall | F1-Score | IoU | mIoU |
|---|---|---|---|---|---|
| Proposed | 77.09 (↓**1.15**) | 81.70 (↑**3.99**) | 79.33 (↑**1.36**) | 65.74 (↑**1.84**) | 80.23 (↑**0.99**) |
| ResU-Net | 78.24 | 77.71 | 77.97 | 63.90 | 79.24 |

To further explicitly evaluate the segmentation potential of the two models, we randomly selected five images to show the mapping results. By visual inspection, it appears that the proposed model outperformed ResU-Net in both datasets. Figure 7 indicated that both models detected the rough boundary of the landslides in the Bijie dataset, while the results of ResU-Net were easily affected by the landslide-like area or the bare soil area, such as Figure 7(aB,aC). Besides, the proposed model successfully discriminated the vegetation area on the landslide body while ResU-Net filtered the vegetation area on the body in Figure 7(aA). For the Iburi dataset, both models had defects on account of the complexity and variety of landslide shape. As shown in Figure 7(bB,bC), the models obtained poor results due to the complicated shapes of the landslides and the varied pixel values on the landslide body. Nevertheless, the proposed model identified more landslide pixels.

### 4.2. Impact of Data Augmentation and CBAM

In this section, we analyzed the impact of data augmentation and CBAM on the proposed model. Firstly, we repeated the first set of experiments without augmentation. In the experiment, the batch size was set to 12 considering the data volume was reduced by seven times. Figure 8 showed the results before and after applying the augmentation strategy, here we analyzed the variation in IoU and mIoU.

For the Bijie dataset, Figure 8a,b indicated that the mIoU has increased by 0.52% and IoU has increased by 0.83% after the augmentation in the proposed model, while the corresponding values were 0.99% and 1.68% in ResU-Net. In the Iburi dataset, the proposed model improved mIoU by 0.52% and IoU by 1.1%, while ResU-Net has im-proved by 0.62% and 1.15% as shown in Figure 8c,d. Compared with the Bijie dataset, the impact on the Iburi dataset was slightly less significant attributed to the distribution characteristics of landslides in the sample image as shown in Figure 6. Therefore, the flip and rotation may have little effect on the dataset. Overall, the promotion contributed by the augmentation strategy on the proposed model was lower than that on the pure FCN.

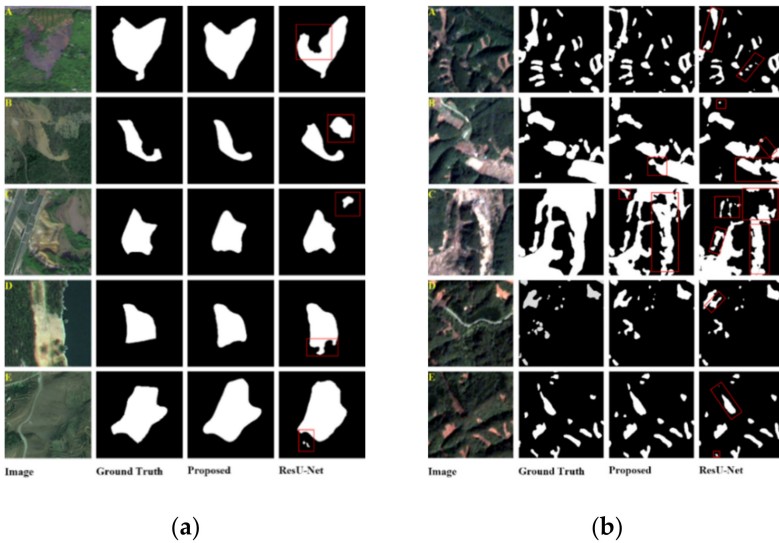

**Figure 7.** The result comparison of two models in the Bijie dataset (**a**) and the Iburi dataset (**b**). Black color represents the background, white color represents the landslide area. Red polygons show the areas that were not fully identified by the two models. A–D are images randomly selected from the testing samples for visual comparison.

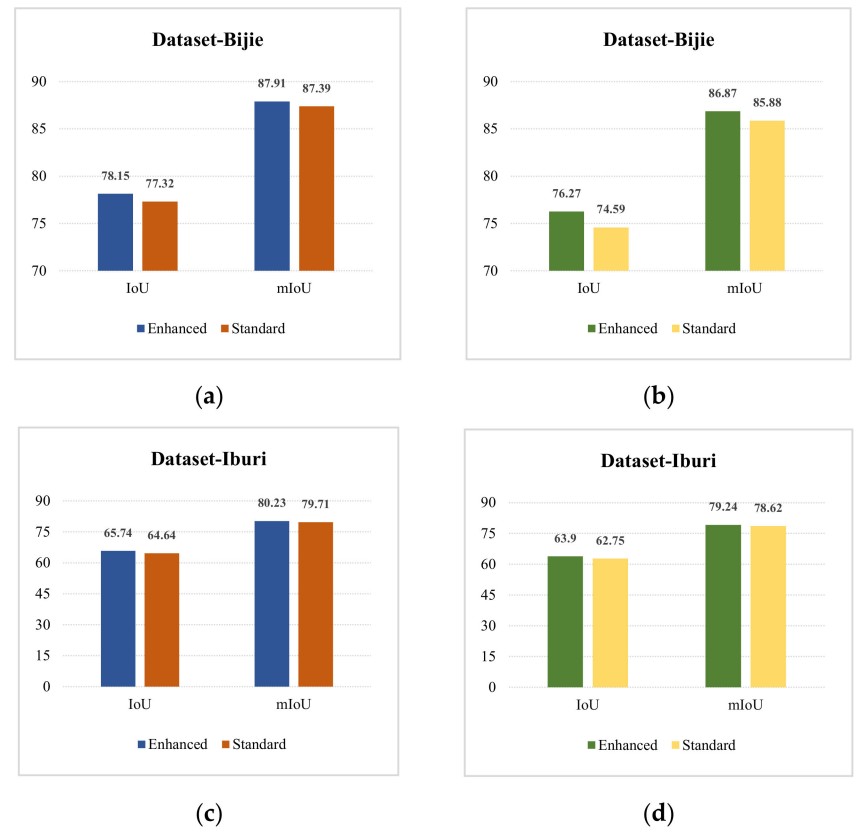

**Figure 8.** The performance of two models on the Bijie and Iburi datasets before and after data augmentation. (**a**,**c**) represent the proposed model, (**b**,**d**) represent ResU-Net.

In theory, the addition of CBAM can filter the features that are unfavorable to the segmentation results and increase the weight of the interesting features to enhance the segmentation performance. Hence, we compared the results with and without the addition of CBAM. The experiments were conducted based on the first set of experiments with only the CBAM module removed. In Figure 9, the model with CBAM has improved in

F1-score, IoU, and mIoU on two datasets to some extent. Specifically for the Bijie dataset, the attention module had less of an increase in results due to the fact that most of the landslides were single with obvious boundaries in the image. By comparison, for the Iburi dataset, the model exhibited a higher enhancement with a 2.27% increase in IoU. As shown in Figure 10(bA,bB), the model with CBAM could identify landslides with complex and fragmented morphology, proving that CBAM indeed assisted the network in better handling landslide information from multiple feature extraction layers. Additionally, the proposed model was more moderate in dealing with the area with apparent color differences on the landslide surface, maximizing the range of the landslide body.

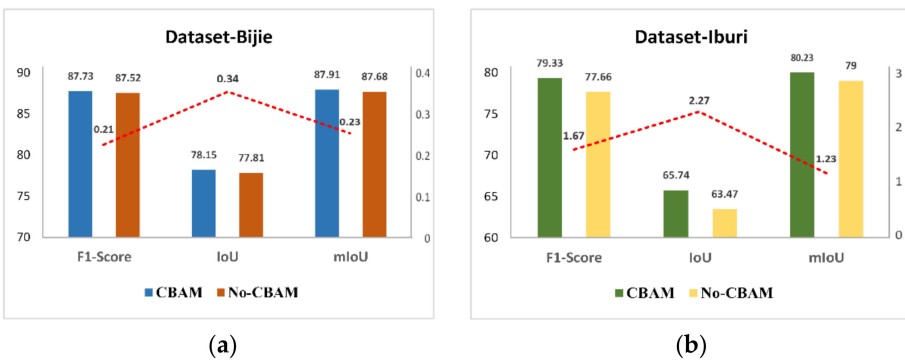

(a) (b)

**Figure 9.** The performance and variation of the proposed model before and after embedding CBAM. (**a**) represents the Bijie dataset, (**b**) represents the Iburi dataset. The dashed line represents the increment after embedding CBAM.

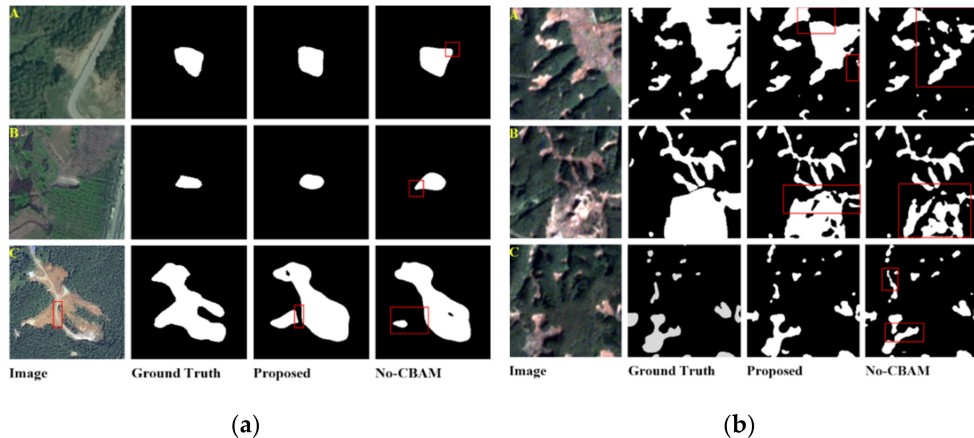

(a) (b)

**Figure 10.** The resulting comparison of two models with and without CBAM in the Bijie dataset (**a**) and the Iburi dataset (**b**). Black color represents the background, white color represents the landslide area. Red polygons show the areas that were not fully identified by the models. A–C are images randomly selected from the testing samples for visual comparison.

## 5. Discussion

This study aimed to explore the potential of a transformer in landslide detection. As a newly proposed approach for image feature extraction, the unique attention mechanism in a transformer allows it to model long-range dependencies for global information on the feature map, while the inductive bias of convolutional networks can regulate the network, ensuring that the network can learn strong representations [53]. By combining CNN and a transformer, the performance of the model can be improved. Besides, it has been demonstrated that fine-tuning the downstream model by introducing the pre-trained transformer weight can accelerate the convergence, which compensates for the premise that a transformer requires large datasets to alleviate weak inductive bias [50]. Additionally, we

embedded a spatial and channel attention module in the decoder to better fuse the features extracted through convolution and the transformer. Therefore, we designed a semantic segmentation network called CTransUnet for landslide extraction, in which ResU-Net was chosen as the base framework with a transformer and CBAM embedded into the encoder and decoder respectively. Furthermore, we tested the model performance using two datasets with different landslide characteristics.

### 5.1. Result Analysis

As shown in Section 4.1, both models obtained relatively higher IoU and F1-scores on different datasets, while the proposed model had a certain degree of improvement for all indicators compared to the pure convolution-based network, proving the feasibility and superiority of transformers. Note that the precision of the proposed model dropped slightly in the Ibrui dataset. To further analyze the reason, the statistics of the confusion matrix were compiled. As shown in Table 5, the proposed model classified more pixels into TP, even though it inevitably misclassified more FP. On the other hand, from the curves of recall and precision [71], the increase of precision may lead to the decrease of recall. Finally, we consider that the increased TP with the cost of slightly higher FP is acceptable for emergency rescue [37,72]. For example, we prefer the pro-posed model to detect more landslide pixels without significantly increasing the number of FN pixels (prefer higher recall rather than higher precision) when generating the distribution map of earthquake-triggered landslides. Although the proposed model classifies more bare soil, building, and road pixels as landslides (lower precision), it can be mitigated by overlaying curvature, slope, and DEM.

**Table 5.** Confusion matrix of two models on the Iburi dataset, ResU-Net (up), proposed (down).

|  | **Prediction False** | **Prediction Truth** |
|---|---|---|
| **Ground False** | 2,684,938 | 75,702 |
| **Ground Truth** | 78,089 | 272,183 |
| **Ground False** | 2,675,614 | 85,026 |
| **Ground Truth** | 64,094 | 286,178 |

In Section 4.2, we found that even the incorporation of a transformer has led to the increased depth and complexity of the network, the proposed model also possessed a higher performance with a small dataset (e.g., only 237 images in the Iburi dataset) by introducing the pre-trained weight compared to the FCN which obviously increased the IoU after being trained with a large dataset. This is slightly different from the view that a transformer relies more on datasets to attenuate the effect of weak inductive bias [49,50]. According to a preliminary analysis, it is mainly because a transformer is not directly used for feature extraction but combined with CNN to better extract global and local semantic information of the feature maps, which is consistent with the findings in [53].

On the other hand, it can be seen from Figure 9 that the indicators of both landslide datasets have improved with the CBAM embedded, and it is worth noting that the proposed model had a higher improvement on the Iburi dataset. Such phenomenon can be ascribed to the difference of the two datasets. Unlike the Bijie dataset, the Iburi dataset suffered from noisy, low-resolution, and fragmented landslide images. However, the addition of CBAM helped the model pay better attention to important features as well as reducing the noise interference, allowing the model to handle landslide areas with large variations in pixel value.

### 5.2. Complexity Comparison of Models

As a landslide detection method aimed at providing auxiliary scientific data for emergency rescue, its complexity and time consumption need to be paid attention. Hence, the trade-offs between the performance and cost must be considered for practical application,

considering that the complex algorithms tend to overload the hard-wires. To evaluate the complexity of the CTransUnet, the number of parameters and training time were compared. As can be seen from Table 6, the costs of introducing transformer into ResU-Net to improve performance are the increase in parameter volume and time consumption.

**Table 6.** The comparison of models on time consumption and parameters.

| Model | Parameter | Training Time |
|---|---|---|
| proposed-Layer12 | 105.4 M | 1.68 h |
| proposed-Layer2 | 34.6 M | 1.25 h |
| ResU-Net | 32.5 M | 0.87 h |

It is worth mentioning that since a transformer is a relatively large part of the proposed model, we reduce the layers of the transformer from standard 12 to 2 to compute its complexity. Table 6 indicated that when the number of layers of the transformer was reduced to two, it has a similar parameters volume as ResU-Net, nevertheless, the proposed model still cost much time to train. The main reason may be that the transformer may have a negative effect on model operations. To further analyze the proposed model performance with two layers, we repeated the first set of experiments on CTransUnet with two layers. As shown in Table 7, when we reduced the layers to two to ensure similar parameter volume with ResU-Net, the performance of the proposed model was still higher than that of the benchmark. Overall, CTransUnet achieved a relative trade-off between the model performance and complexity.

**Table 7.** The performance of the proposed model with transformer Layer = 2. The bold means the improvement in the proposed model.

| | Model | F1-Score | IoU | mIoU |
|---|---|---|---|---|
| BiJie | proposed-layer2 | 87.54 (↑**1.01**) | 77.84 (↑**1.57**) | 87.75 (↑**0.88**) |
| | ResU-Net | 86.53 | 76.27 | 86.87 |
| Ibrui | proposed-layer2 | 78.22 (↑**0.25**) | 64.23 (↑**0.33**) | 79.45 (↑**0.21**) |
| | ResU-Net | 77.97 | 63.90 | 79.24 |

*5.3. Deficiencies of Experiments*

Although transformers are gradually becoming a popular and powerful approach in computer vision because of their unique attention mechanism and weak inductive bias, in a majority of related works focused on how to transplant transformers to classification and segmentation, fewer researchers explored the mechanisms between transformers and CNN. In addition, there are still some aspects not considered in this study.

- How to effectively utilize and optimize the feature maps generated from CNN for the transformer, avoiding the loss of global contextual information.
- How to reduce the reliance of models on pre-trained weights with a small training dataset, considering that the specialized images are different from the public datasets.
- The efficient network architecture was not explored comprehensively. For instance, the position of the transformer in ResU-Net.
- In order to match the pre-trained weights, the image size used in the model was fixed. Hence, multi-source remote sensing images were not taken into consideration for extracting more abundant information.
- The number of samples needs to be quantitatively analyzed to evaluate its impact on the results.

To fully grasp the mechanism of transformers and their strength when fused with the CNN model, the above issues need to be explored in future research thoroughly, contributing to more flexible integration of transformers with various powerful CNN-based models. It has been demonstrated that a specific loss function can help ViT to outperform ResNet without data augmentation and pre-trained weights [73].

## 6. Conclusions

In this study, we first proposed a new DL-based approach denoted as CTransUnet for landslide detection with small training samples. It combines the strength of two sub-networks, namely ResU-Net and a transformer. The former was used to extract the local and sophisticated features while the latter was introduced to model the long-range dependence in the spatial relationships of the feature maps. Besides, a spatial-channel attention module (CBAM) was embedded into skip connection of the proposed model to increase the weight values of feature maps that were more important for segmentation. Finally, the pre-trained weights were introduced to accelerate the convergence. To evaluate the performance quantitatively, standard ResU-Net and five criteria were selected in experiments. The results showed that CTransUnet possessed optimum performance on both datasets with an IoU 65.74% and 78.15%. In addition, we explored the impact of data augmentation and CBAM on the proposed model. It was prominent that CTransUnet possessed satisfying results even without data augmentation and CBAM had the capability in optimizing the model to focus on the important features and process complex landslide morphologies. Overall, this study demonstrated the outstanding potential of transformers and CNN in landslide extraction, which can provide a scientific data source for emergency response and landslide susceptibility mapping. However, the transformer-based landslide detection approaches are still in infancy, the proposed issues in Section 5.3 should be investigate in further study.

**Author Contributions:** C.X. proposed the research concept, organized landslide interpretation work, and offered basic data. L.L. participated in the landslide interpretation and data preparation work. Z.Y. designed the framework of the research, conducted the experiments, and wrote the manuscript. All authors have read and agreed to the published version of the manuscript.

**Funding:** This research was supported by the National Institute of Natural Hazards, Ministry of Emergency Management of China (ZDJ2021-14), the Lhasa National Geophysical Observation and Research Station (NORSLS20-07), and the National Key Research and Development Program of China (2018YFC1504703).

**Acknowledgments:** We are grateful to the editors and anonymous reviewers for their constructive comments and suggestions that improved the quality of the manuscript.

**Conflicts of Interest:** The authors declare that there are no conflict of interest.

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
