# Peer review of "Landslide Detection Based on ResU-Net with Transformer and CBAM Embedded: Two Examples with Geologically Different Environments"

_remotesensing, doi:10.3390/rs14122885_

Round 1
Reviewer 1 Report
This paper integrates vision transformer into CNN to improve the landslide detection results especially for the small dataset. The applicability of the new proposed method is validated with two different geological landslides. This study has distinct innovation for modeling and improvement for real application, which can be widely tested for landslide inventory mapping. Some specific comments are given bellow.
1. Abstract (Line 14-15). ‘…reducing the time consuming for feature engineering’. The description of feature engineering is vague for audience. CNN can extract depth features, and feature engineering mainly deals with shallow features for classification. They are different stages of feature extraction. The author needs to rephrase the statement here.
2. It is emphasized in the abstract that the method can be used for small datasets. How do the author avoid the over-fitting of small datasets?
3. Line 159. Figure 1(c) is not the Bijie-dataset.
4. Line 372. Dice Loss mitigated the sample imbalance. What is the meaning of ζ in two groups of experiments?
5. As described in Section 2.1. The two datasets have different resolutions. ViT is for adding global information, CBAM is for filtering. How does the introduction of these two different modules work on data sets with different resolutions?
6. In Figure 9, the validation parameters before and after CMAM needs to be added for visual comparison.
7. The authors adds ViT after the third convolution block, which may be limited to improving global information sampling. Why don’t add ViT after the first convolution block?
Reviewer 2 Report
In this study, the authors applied deep learning methods to predict the landslide potentials of an area. While the topic is interesting, clarifications and justifications are required regarding the contribution and validation of this study. I would like the authors to address the following comments diligently before the manuscript can be accepted for publication.
· This study primarily focused on promoting deep learning methods for landslide prediction. How does it help improve the scientific knowledge of remote sensing technology? It only uses remotely sensed data. How does the manuscript relevant to the Remote Sensing journal?
· For me "landslide detection" and "landslide prediction" are not the same. The title indicates that this study develops a method for landslide observation from remote sensing data. However, the authors mainly focused on predicting landslide potential areas based on a few sample landslide locations. I would suggest the authors revise the title of this study.
· In relation to deep learning-based landslide prediction, many studies have already been conducted. A few of many examples are given below. The authors must justify how their study improves the prevailing knowledge on this topic. A mere application of a new algorithm is not sufficient to conduct a study.
https://doi.org/10.1016/j.catena.2019.104426
https://doi.org/10.1016/j.catena.2019.104451
https://doi.org/10.3390/rs12030346
· Prediction of landslides using machine learning or deep learning methods could be subject to uncertainties. In the introduction section, the authors should explain various sources of uncertainties regarding such methods. They could consult the following articles to improve the motivation of this study.
https://doi.org/10.3390/rs12203347
https://doi.org/10.1016/j.geomorph.2009.06.020
https://doi.org/10.1016/j.geomorph.2010.09.004
https://doi.org/10.1016/j.geomorph.2006.04.007
· Section 2.1: The authors need to justify the choice of this study area.
· The methodology and results section overlapped on several occasions. For instance, sections 4.1 and 4.2 should be part of the methodology.
· What are the potential sources of uncertainties in the results? How were the uncertainties treated in this study?
Reviewer 3 Report
This paper presents the results of research „ Landslide Detection Based on ResU-Net with Transformer and CBAM Embedded: Two Examples with Geologically Different Environments”
In this paper, authors proposed to integrate transformer into ResU-Net for landslide detection task with small datasets. The results indicated that the proposed model obtained the highest mIoU and F1-score in both datasets, demonstrating that the ResU-Net with transformer embedded can be used as a robust landslide detection method and thus realize the generation of accurate regional landslide inventory and emergency rescue. These are promising results. I hope the authors will continue their research on this issue. I also hope that the results of their work will find application.
More specifically, I have the following main comments:
ü About study area
It is necessary to correct Fig. 1
ü About the Discussion and Conlusion
It is worth emphasizing more clearly what was new about the results of the research (in Conclusion secion).
Finally the chapter 5.3 Future studies - Please move this chapter to the conclusion section
References
The references contains 69 items. This is a sufficient amount. The most recent items are cited
For further detailed comments, see the accompanying manuscript.

Reviewer 4 Report
Dear author,
there is no duplicate in this paper, it is an interesting research. However, please revise those minor mistakes below:
Line 33: please delete "landslide" and move it , put it before "are triggered...". please refer my correction: As one of the most critical types of natural hazards, Landslides are triggered by various external factors in most cases, including earthquakes, rainfall, variation of water level storms, and river erosion [1].
Line 44: the word "huge" is often overuse. please change to "massive".
Line 49, please delete "in general" it is not necessary
Line 50-53: This sentence is unclear, please add "the" before "pixel-based method". Delete the phrase word or re-arrange "which extracts landslides mainly by the comparison of ", change it to "which mainly extracts landslides by comparing"
Line 75: delete "great" it is often overused. change to " excellent"
Line 83: delete combined, it appears repeatedly in this text. Please change to "Connected"
Line 116-120: this sentence is unclear, please slit it into two sentences.
All the equations need to be mentioned in the relevant content.
Line 677-678: check the page, page 5 of 18? Please check the format, this is applying for Remote sensing, not ISPRS.
The other parts are well written.

Round 2
Reviewer 2 Report
I would like to thank the authors to address my comments diligently. I recommend publication of the revised manuscript.
